# Distinctive Expression and Amplification of Genes at 11q13 in Relation to HPV Status with Impact on Survival in Head and Neck Cancer Patients

**DOI:** 10.3390/jcm7120501

**Published:** 2018-12-01

**Authors:** Francisco Hermida-Prado, Sofía T. Menéndez, Pablo Albornoz-Afanasiev, Rocío Granda-Diaz, Saúl Álvarez-Teijeiro, M. Ángeles Villaronga, Eva Allonca, Laura Alonso-Durán, Xavier León, Laia Alemany, Marisa Mena, Nagore del-Rio-Ibisate, Aurora Astudillo, René Rodríguez, Juan P. Rodrigo, Juana M. García-Pedrero

**Affiliations:** 1Department of Otolaryngology, Hospital Universitario Central de Asturias and Instituto de Investigación Sanitaria del Principado de Asturias, Instituto Universitario de Oncología del Principado de Asturias, University of Oviedo, 33011 Oviedo, Spain; franjhermida@gmail.com (F.H.-P.); sofiatirados@gmail.com (S.T.M.); pavlikh@gmail.com (P.A.-A.); rocigd281@gmail.com (R.G.-D.); saul.teijeiro@gmail.com (S.Á.-T.); angelesvillaronga@gmail.com (M.Á.V.); ynkc1@hotmail.com (E.A.); laurita_alonso86@hotmail.com (L.A.-D.); nagoredelrio@gmail.com (N.d.-R.-I.); renerg.finba@gmail.com (R.R.); 2Centro de Investigación Biomédica en Red Cáncer, CIBERONC, 28029 Madrid, Spain; mmena.iconcologia@gmail.com; 3Otorhinolaryngology Department, Hospital de la Santa Creu i Sant Pau, Universitat Autònoma de Barcelona, 08041 Barcelona, Spain; XLeon@santpau.cat; 4Centro de Investigación Biomédica en Red de Bioingeniería, Biomateriales y Nanomedicina, CIBER-BBN, 28029 Madrid, Spain; 5Cancer Epidemiology Research Program, Catalan Institute of Oncology (ICO), IDIBELL. L’Hospitalet de Llobregat, 08908 Barcelona, Spain; alemanyvilchesico@gmail.com; 6Centro de Investigación Biomédica en Red de Epidemiología y Salud Pública, CIBERESP, Barcelona, Spain; 7Department of Pathology, Hospital Universitario Central de Asturias, Instituto Universitario de Oncología del Principado de Asturias, University of Oviedo, 33011 Oviedo, Spain; astudillo@hca.es

**Keywords:** head and neck squamous cell carcinoma, HPV, 11q13, gene amplification, immunohistochemistry

## Abstract

Clear differences have been established between head and neck squamous cell carcinomas (HNSCC) depending on human papillomavirus (HPV) infection status. This study specifically investigated the status of the *CTTN*, *CCND1* and *ANO1* genes mapping at the 11q13 amplicon in relation to the HPV status in HNSCC patients. CTTN, CCND1 and ANO1 protein expression and gene amplification were respectively analyzed by immunohistochemistry and real-time PCR in a homogeneous cohort of 392 surgically treated HNSCC patients. The results were further confirmed using an independent cohort of 279 HNSCC patients from The Cancer Genome Atlas (TCGA). The impact on patient survival was also evaluated. *CTTN*, *CCND1* and *ANO1* gene amplification and protein expression were frequent in HPV-negative tumors, while absent or rare in HPV-positive tumors. Using an independent validation cohort of 279 HNSCC patients, we consistently found that these three genes were frequently co-amplified (28%) and overexpressed (39–46%) in HPV-negative tumors, whereas almost absent in HPV-positive tumors. Remarkably, these alterations (in particular CTTN and ANO1 overexpression) were associated with poor prognosis. Taken together, the distinctive expression and amplification of these genes could cooperatively contribute to the differences in prognosis and clinical outcome between HPV-positive and HPV-negative tumors. These findings could serve as the basis to design more personalized therapeutic strategies for HNSCC patients.

## 1. Introduction

Head and neck cancer represents a heterogeneous group of tumors that accounts for about 5% of the total annual worldwide cases of cancer, usually associated with poor prognosis [1,2]; however, its incidence and other main features can significantly vary from one region to another. The most prevalent histological type in head and neck cancers is the squamous cell carcinoma (HNSCC), and it is widely accepted that the most important risk factors are tobacco and alcohol consumption [1,3,4]. Interestingly, the incidence of HNSCCs is currently declining in some specific regions, due to a decrease of tobacco and alcohol consumption [4,5]. Nevertheless, there is an increasing incidence of oropharyngeal tumors that are associated with human papillomavirus (HPV) infection in certain developed countries [4,5].

The proportion of HPV-related oropharyngeal tumors can vary considerably among regions; the observed prevalence in Europe varies from an estimated 17% in southern countries to as high as 93% in northern countries like Sweden [5]. More than 90% of HPV-related HNSCC tumors are caused by one specific virus type, the HPV-16, which also leads to HPV-related anogenital tumors [4,5]. On this basis, epidemiological studies have correlated HPV-positive tumors with sexual behaviors, suggesting that the risk increases with the number of sexual partners and is particularly more frequent in men [4,5].

Recent studies have consistently showed that HPV-related (HPV+) and unrelated (HPV−) tumors represent two different entities in terms of clinical, biological, and molecular characteristics. Thus, HPV-positive tumors generally arise in the oropharynx, have a more favorable prognosis, a better response to chemotherapy and radiotherapy, and less genetic alterations when compared to HPV-negative tumors [2,4,5,6].

One of the most frequent genetic alterations found in HNSCCs is the amplification of the chromosomal region 11q13, which has been associated with tumor aggressive behavior, increased lymph node metastases, and poor outcome [7]. Moreover, the comprehensive molecular analysis of the 11q13 region led to the identification of genes evidenced to be directly related to an improved ability for tumor growth, migration and invasion. Some of the genes mapping to the 11q13 region, are *CTTN* (Cortactin), *CCND1* (Cyclin D1), and the more recently discovered *ANO1* (Anoctamin-1). *CTTN* and *CCND1* are well-established oncogenes, associated with advanced disease stage and poor prognosis [8,9]. On the other hand, the newly characterized *ANO1* encodes a calcium-activated transmembrane chloride channel whose overexpression and amplification have been correlated with increased cell migration and propensity to develop metastases [10,11]. However, data revealing the role and importance of *ANO1* in HNSCC have only recently emerged and, hence, are still limited.

Recent studies have evidenced important differences in the molecular alterations and the impact of the 11q13 chromosomal region in HNSCC, depending on the anatomic site of the tumor and the HPV infection status. These studies suggest that the amplification of 11q13 in HNSCCs is more frequent in HPV-negative than in HPV-positive tumors [12], and this presumably contributes to the better prognosis of the latter. On the basis of the increasing incidence of HPV-related tumors, the current information regarding the clinical differences between HPV-related and unrelated tumors, and the discovery of promising molecular prognosis factors like *ANO1*, we believe it is crucial to study in depth the molecular alterations involved in HNSCC and their relationship with HPV infection, in order to have a better understanding of the pathogenesis of the disease with the ultimate goal of establishing the path for the development of new and more effective treatment strategies. This consideration prompted us to study the status of various genes (*CTTN*, *CCND1* and *ANO1*) mapping at the 11q13 amplicon in relation to HPV infection in two large independent cohorts of HNSCC patients.

## 2. Materials and Methods

### 2.1. Patients and Tissue Specimens

Surgical tissue specimens from 392 patients with HNSCC who underwent resection of their tumors at the Hospital Universitario Central de Asturias and Hospital Sant Pau between 1990 and 2010 were retrospectively collected, in accordance to approved institutional review board guidelines. All experimental protocols were conducted in accordance to the Declaration of Helsinki and approved by the Institutional Ethics Committees of the Hospital Universitario Central de Asturias and Hospital Sant Pau and by the Regional CEIC from Principado de Asturias (approval number: 81/2013 for the project PI13/00259). Informed consent was obtained from all patients. Representative tissue sections were obtained from archival, paraffin-embedded blocks, and the histological diagnoses were confirmed by an experienced pathologist (Aurora Astudillo). 

Clinical, sociodemographic, follow-up, and risk factors information was collected from the medical records. The stage of the tumors was determined according to the TNM system of the International Union Against Cancer (7th Edition). 

### 2.2. Tissue Microarray (TMA) Construction and DNA Extraction

The original archived hematoxylin- and eosin-stained slides were reviewed by an experienced pathologist (Aurora Astudillo) to confirm the histological diagnoses. Five morphologically representative areas were selected from each individual tumor paraffin block: two for DNA isolation and three for the construction of a TMA. To avoid cross-contamination, two punches of 2 mm diameter were taken first, using a new, sterile punch (Kai Europe GmbH, Solingen, Germany) for every tissue block, and stored in eppendorf tubes (Sigma Aldrich, Saint Louis, MO, USA) at room temperature. Subsequently, three representative tissue cores (1 mm diameter punches) were selected from each tumor block and transferred to a recipient ‘Master’ block in a grid-like manner using a manual tissue microarray instrument to construct TMA blocks, as described previously [13,14]. In addition, each TMA included three cores of normal epithelium (tonsil) as an internal negative control and three cores of a HPV-positive cervix carcinoma as a positive control. A section from each microarray was stained with hematoxylin and eosin and examined by light microscopy (Leica Microsystems, Wetzlar, Germany) to check the adequacy of tissue sampling.

The protocol for DNA extraction from paraffin-embedded tissue sections has been described elsewhere [13]. Briefly, Formalin-fixed paraffin-embedded (FFPE) tissue samples were subjected to thorough deparaffinization with xylene (Sigma Aldrich, Saint Louis, MO, USA), methanol (Merck, Darmstadt, Germany) washings to remove all traces of the xylene, and a 24 h incubation in 1 mol/L sodium thiocyanate (Sigma Aldrich Inc.) to reduce cross-links. Subsequently, the tissue pellet was digested for 2–3 days in lysis buffer with high doses of proteinase K (final concentration, 2 µg/µL, freshly added twice daily). Finally, DNA extraction was done using the QIAamp DNA Mini Kit (Qiagen GmbH, Hilden, Germany).

### 2.3. HPV Detection

The algorithm used to detect the presence of HPV in these patients has been previously described in detail [13,15]. Briefly, the presence of HPV was assessed by p16-immunohistochemistry in all cases, and those cases showing p16-positive immunostaining (any nuclear and or cytoplasmic staining) were subjected to high-risk HPV DNA detection and genotyping by GP5+/6+-PCR with an enzyme-immuno-assay (EIA) read-out for detection of 14 high-risk HPV types. Subsequent genotyping of EIA-positive cases was performed by bead-based array on the Luminex platform. In addition, in situ hybridization (ISH) with biotinylated HPV DNA probes considered to react with HPV types 16, 18, 31, 33, 35, 39, 45, 51, 52, 56, 58, 59, and 68 (Y1443, DakoCytomation, Glostrup, Denmark) was performed on all carcinomas, using 3 µm tissue sections of the TMAs, according to the manufacturer’s instructions.

### 2.4. Gene Amplification Analysis

Gene amplification was evaluated by real-time PCR (Q-PCR) in an ABI PRISM 7500 Sequence detector (Applied Biosystems, Foster City, CA) using Power SYBR Green PCR Master Mix and oligonucleotides designed according to Primer Express software v2.0 (Applied Biosystems, Lincoln Centre Drive Foster City, CA, USA) with the following sequences: for *CCND1* gene, Fw, 5′-GGAAGATCGTCGCCACCTG-3′ and Rv, 5′- GAAACGTGGGTCTGGGCAAC-3′; for *ANO1* gene, Fw, 5′-CAAAGGCAGGTGCTTTGCA-3′ and Rv, 5′-TCTACGGGCCTCTGCTCACT-3′; for *CTTN* gene, Fw, 5′-GATCTCATTTGACCCTGATGACATC-3′ and Rv, 5′-CGTACCGGCCCTTGCA-3′; and for the reference gene *TH* (Tyrosine Hydroxylase, located at 11p15), Fw, 5′-TGAGATTCGGGCCTTCGA-3′ and Rv, 5′-GACACGAAGTAGACTGACTGGTACGT-3′. Dissociation curve analysis of all PCR products showed a single sharp peak, and the correct size of each amplified product was confirmed by agarose gel electrophoresis. The relative gene copy number was calculated using the 2^-ΔΔ*C*T^ method, as we previously described [16,17]. The ΔΔ*C*_T_ represents the difference between the paired tissue samples (Δ*C*_T_ of tumor − Δ*C*_T_ of normal mucosa), with Δ*C*_T_ being the average *C*_T_ for each target gene minus the average *C*_T_ for the reference gene (*TH*). Relative copy numbers ≥2-fold were considered gene gain, and relative copy numbers ≥4-fold were considered positive for gene amplification.

### 2.5. Immunohistochemistry

The formalin-fixed, paraffin-embedded tissues were cut into 3-µm sections and dried on Flex IHC microscope slides (Dako, Glostrup, Denmark). The sections were deparaffinized with standard xylene and hydrated through graded alcohols into water. Antigen retrieval was performed using Envision Flex Target Retrieval solution, high pH (Dako, Glostrup, Denmark). Staining was done at room temperature on an automatic staining workstation (Dako Autostainer Plus) with mouse anti-Cyclin D1 monoclonal antibody DCS-6 (Santa Cruz Biotechnology, Inc. sc-20044) at 1:100 dilution, rabbit polyclonal Anti-TMEM16A antibody (Abcam # ab53212) at 1:500 dilution, or mouse anti-cortactin monoclonal antibody Clone 30 (BD Biosciences Pharmingen, San Jose, CA, USA) at 1:200 dilution, using the Dako EnVision Flex + Visualization System (Dako Autostainer). Counterstaining with hematoxylin was the final step. 

The slides were viewed randomly without clinical data by two of the authors, according to the scoring systems previously described [17,18] with a high level of inter-observer concordance (>95%). CCND1 staining was evaluated as the percentage of cells with nuclear staining, scored as 0–2 according to the semiquantitative scale (0–10, 10–50, or >50% positive tumor cells). CCND1 staining scores were classified as negative or positive staining on the basis of values below or above the median value of 10%. Since CTTN and ANO1 staining showed a homogeneous distribution, a semiquantitative scoring system based on staining intensity was applied. Thus, CTTN immunostaining was scored as negative (0), weak (1), moderate (2), and strong protein expression (3). The staining data were dichotomized as negative (scores 0 and 1) versus positive (scores 2 and 3). ANO1 immunostaining was scored as negative (0), weak to moderate (1), and strong protein expression (2). Scores ≥1 were considered positive ANO1 expression. 

### 2.6. Statistical Analysis

Fisher’s exact test was used for comparison between categorical variables, and t Student test for parametric continuous variables. Correlations between protein expression, gene amplification, and HPV status were estimated by Kendall‘s tau correlation test. All tests were two-sided; *p* values ≤0.05 were considered to be statistically significant. 

## 3. Results

### 3.1. Patient Characteristics

A large cohort of 392 homogeneous surgically treated HNSCC patients was selected for study. A flow diagram of the experimental setup is shown in Appendix A. All patients had a single primary tumor, microscopically clear surgical margins, and received no treatment prior to surgery. Only 19 patients were women, and the mean age was 60 years (range 30 to 89 years). All but 22 patients were habitual tobacco smokers—202 moderate (1–50 pack-year) and 159 heavy (>50 pack-year)—and 353 were alcohol drinkers. Nineteen tumors were stage I, 25 stage II, 69 stage III, and 279 stage IV. The series included 152 tonsillar, 116 base of tongue, 62 hypopharyngeal, and 62 laryngeal carcinomas. A total of 147 tumors were classified as well-differentiated, 151 as moderately differentiated, and 94 as poorly differentiated. In total, 232 (59%) patients received postoperative radiotherapy. 

### 3.2. Distinctive Associations of CCND1, ANO1, and CTTN Protein Expression with HPV Status in HNSCC Patients

We found that 67 cases (17%) showed nuclear and cytoplasmic p16 expression, a surrogate marker for HPV infection (2) (Table 1). HPV DNA was assessed by GP5+/6+-PCR and by in situ hybridization in the 67 p16-positive cases, which resulted in a total of 30 HPV-positive cases (28 oropharyngeal, 1 laryngeal, and 1 hypopharyngeal carcinoma) in our series (all were HPV type 16). Representative images of HPV-positive and HPV-negative cases and also examples of p16 immunostaining are shown in Figure 1. We found a total of 267 (68%) positive cases for CCND1 protein expression, 78 (21%) positive cases for ANO1 expression, and 190 (49%) positive cases for CTTN expression (Table 1). Representative examples of protein staining are shown in Figure 2, and raw data in Appendix A. We next investigated the relationship between CCND1, ANO1, and CTTN expression and HPV infection status. The expression of these three proteins was strongly and inversely correlated with HPV infection. Notably, all the HPV-positive cases showed negative ANO1 expression, and 28 HPV-positive cases had also negative CTTN expression (Table 1). 

### 3.3. Analysis of CCND1, ANO1, and CTTN Gene Amplification in Relation to HPV Status in HNSCC Patients

*CCND1*, *ANO1*, and *CTTN* gene gain and amplification was assessed by real-time PCR in 88 cases selected from the same HNSCC tissue blocks, including 26 HPV-positive cases. We found gene gains and amplifications of *CCND1* in 32 (36%) tumors, *ANO1* in 42 (48%) tumors, and *CTTN* in 26 (30%) tumors (Table 2), with relative gene copy numbers ranging from 2- to 24-fold. Co-amplifications of these three genes were also frequently observed (26 cases, 30%). We also found that tumors harboring gene amplification significantly correlated with higher expression levels of each protein (Figure 3). When analyzing the correlations with HPV infection status, we consistently found that *CCND1*, *ANO1*, and *CTTN* gene amplification inversely correlated with HPV status (Table 2). Thus, amplifications of *CTTN, CCND1*, and *ANO1* were frequent in HPV-negative tumors (ranging from 42 to 61%), while absent in HPV-positive tumors.

### 3.4. Analysis of CCND1, ANO1, and CTTN mRNA Expression in Relation to HPV Status in 279 HNSCC Patients from the TGCA

In order to confirm our results, we performed an analysis of the publicly available TCGA data from 279 HNSCC patients [19] using the platform cBioPortal for Cancer Genomics (http://cbioportal.org/) [20]. The clinicopathologic characteristics of this cohort are summarized in Appendix A and Appendix A. A total of 36 (13%) patients were positive for HPV infection, most prevalent in the oropharynx (22 cases, 61%), and associated to lower tobacco consumption, lower mutations, improved survival, and a younger age in both men and women. The results also evidenced differential changes in mRNA expression levels of *CTTN*, *CCND1* and *ANO1* depending on HPV infection status. Thus, increased mRNA expression levels of these genes were frequently and significantly observed in HPV-negative tumors (Figure 4A–D), while very rare in HPV-positive patients (Table 3). Similarly, the analysis of copy number alterations of *CTTN, CCND1*, and *ANO1* genes also revealed that both amplifications and gains of these three genes were also highly frequent in HPV-negative tumors, whereas almost absent in HPV-positive tumors (Figure 4A,E). Furthermore, the results consistently showed that these three genes were frequently co-amplified (28%) and overexpressed at a higher frequency (39–46%), and, more importantly, these molecular alterations (in particular CTTN and ANO1 overexpression) were associated with a worse clinical outcome in the TCGA cohort of 279 HNSCC patients and also in an extended TCGA cohort which added 251 new HNSCC patients (*n* = 530) (Figure 5).

## 4. Discussion

The incidence of HPV-related HNSCC is currently increasing and gaining importance, while tumors associated to tobacco and alcohol consumption are declining [4,5]. Recent studies have established important differences between HNSCC depending on HPV infection status, including clinical, biological, and molecular features, emphasizing the presence of less genetic alterations, a better response to chemotherapy and radiotherapy, and a more favorable prognosis for HPV-related HNSCC [2,4,5]. These differences could change the way we diagnose, treat, and manage HNSCC. One of the most frequent genetic alterations found in HNSCC is the amplification of the 11q13 locus, which has been associated with increased tumor growth, proliferation, and dissemination [8,9]. 

Given the importance of the 11q13 locus in HNSCC and to contribute to the molecular characterization of these tumors, we conducted a study on a large unbiased cohort of 392 homogeneous surgically treated HNSCC patients to investigate the role of 11q13 amplification in relation to HPV status. This was accomplished by assessing the specific relationship of the protein expression and amplification of the *CTTN, CCND1*, and *ANO1* genes mapping within this locus. Immunohistochemical analysis of CCND1, CTTN, and ANO1 revealed that the expression of these three proteins was strongly and inversely correlated with HPV infection. Noteworthy, all the HPV-positive cases showed negative ANO1 expression, and 28 out of 30 HPV-positive cases had also negative CTTN expression. Likewise, the analysis of gene copy amplification by real-time PCR consistently showed that amplifications of *CTTN*, *CCND1* and *ANO1* were frequently detected in HPV-negative tumors (ranging from 42 to 61%), while absent or rare in HPV-positive tumors (0–15%). Mechanistically, we found that the contribution of gene amplification to protein expression varied widely depending on each gene. Even though tumors harboring amplification also concomitantly expressed high protein levels, positive CCND1 and CTTN expression occurred at a higher frequency than gene amplification, while ANO1 protein expression was less frequent than *ANO1* gene amplification. Additional regulatory mechanisms (transcriptional and post-translational) should contribute to protein expression, as previously reported [18]. A limitation of our study is the use of tissue microarrays to evaluate protein expression, which may constitute a drawback to assess the possible influence of tumor heterogeneity. To minimize this, three different representative tumor areas were selected from each tissue block and analyzed in the TMAs. Of note, the results showed that these proteins presented homogeneous and highly concordant expression patterns in the three tissue punches from each tumor.

These results were further and significantly validated using an independent cohort of 279 TCGA HNSCC patients, thus confirming that the mRNA expression levels of CCND1, CTTN, and ANO1 were significantly increased in a high proportion of HPV-negative tumors, whereas most HPV-positive patients exhibited normal mRNA levels of all these genes. Similar observations were obtained by analyzing the frequencies of copy number alterations. These findings are also consistent with some studies that reported that 11q13 amplifications and gains were less frequent in HPV-related tumors [2,4]. This is also in agreement with the assumption made by Kostareli *et al.* [6] that, in HPV-positive tumors, a lower amount of genetic alterations is required to achieve the malignant phenotype, because of the inactivation of p53 and pRb proteins by the viral E6 and E7 oncoproteins. To our knowledge, this is the first study to assess specifically the protein expression and copy number alterations of various genes mapping at the 11q13 amplicon in relation to HPV infection status using two large independent cohorts of HNSCC patients. Our study unveils important differences regarding the expression and amplification of the *CCND1*, *CTTN* and *ANO1* genes between HPV-related and unrelated tumors. It is worth mentioning that the two HNSCC cohorts selected for study are representative, sharing multiple of the unique characteristics reported for HPV-positive tumors [13], such as a clear prevalence in the oropharynx, low tobacco and alcohol consumption by the patients, lower tumor stage, basaloid histological pattern, a younger patients’ age at diagnosis, lower mutations and CNA, and above all, a better prognosis. A recent study by Dixit *et al.* [21] provided the first evidence of a link between ANO1 expression and gene amplification and HPV status using a series of 64 pharyngeal tumors and tissue microarrays for IHC evaluation. Therefore, our results further and significantly strength and validate these preliminary findings on ANO1 protein expression using a large independent cohort of 392 HNSCC patients, as well as on ANO1 mRNA levels in the TCGA cohort of 530 HNSCC patients.

Together, these findings suggest that the *CCND1*, *CTTN* and *ANO1* genes within the 11q13 amplicon could play a significant role in HPV-negative tumors but not in HPV-positive tumors. Given that 11q13 amplification has been associated with poor prognosis in HNSCC patients, and, in particular, these three genes have been involved in tumor progression and resistance to radio-, chemotherapy [11,22,23,24,25,26], and EGFR-targeted therapies [27], the herein found distinctive molecular alterations presumably could contribute to the clinical and biological differences between these two different HNSCC subtypes and explain the better prognosis and response to radiotherapy and chemotherapy associated to HPV-positive tumors. In fact, we proved the impact of these molecular alterations on patient survival. In particular, CTTN and ANO1 overexpression, rather than gene amplification, was more frequent and found to associate with a worse clinical outcome in two TCGA cohorts of 279 and 530 HNSCC patients. Nevertheless, given the size of the 11q13 amplicon, it cannot be ruled out that the overexpression of these genes may be secondary and that other genes within the 11q13 amplicon could act as true drivers of HNSCC progression. A recent study has also identified four genes (*ORAOV1*, *CPT1A*, *SHANK2* and *PPFIA1*) as important drivers of 11q13 amplification that showed an impact on prognosis [28]. Therefore, various genes within the 11q13 amplicon could cooperatively contribute to the differences in prognosis and clinical outcome between HPV-positive and HPV-negative tumors. In summary, various studies including ours have provided strong evidence demonstrating that HPV-related and unrelated tumors are two different entities; hence, we consider that the management of HNSCC patients should move toward a more personalized approach.

## 5. Conclusions

*CTTN*, *CCND1* and *ANO1* amplification and overexpression were frequent in HPV-negative tumors and correlated with reduced patient survival, while absent or very rare in HPV-positive tumors. Therefore, these molecular alterations could contribute to the distinct clinical outcomes of these two HNSCC entities and serve as the basis to design more personalized therapeutic strategies for HNSCC patients.

## Figures and Tables

**Figure 1 jcm-07-00501-f001:**
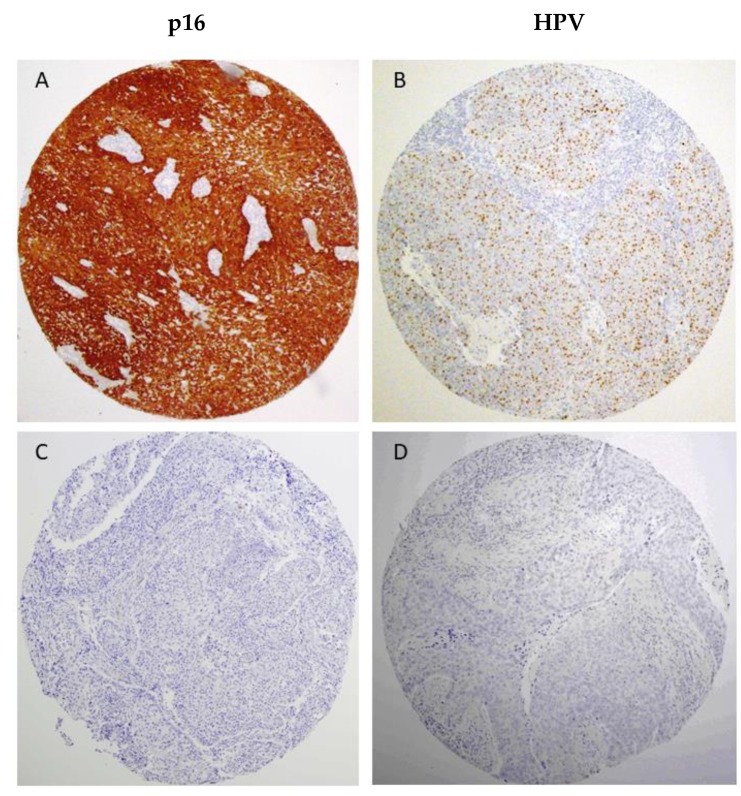
Representative images of p16-positive (**A**) and negative (**C**) immunostaining in the HNSCC tissue microarrays (TMAs) and HPV-positive (**B**) and HPV-negative (**D**) cases detected by in situ hybridization. Original magnification ×10.

**Figure 2 jcm-07-00501-f002:**
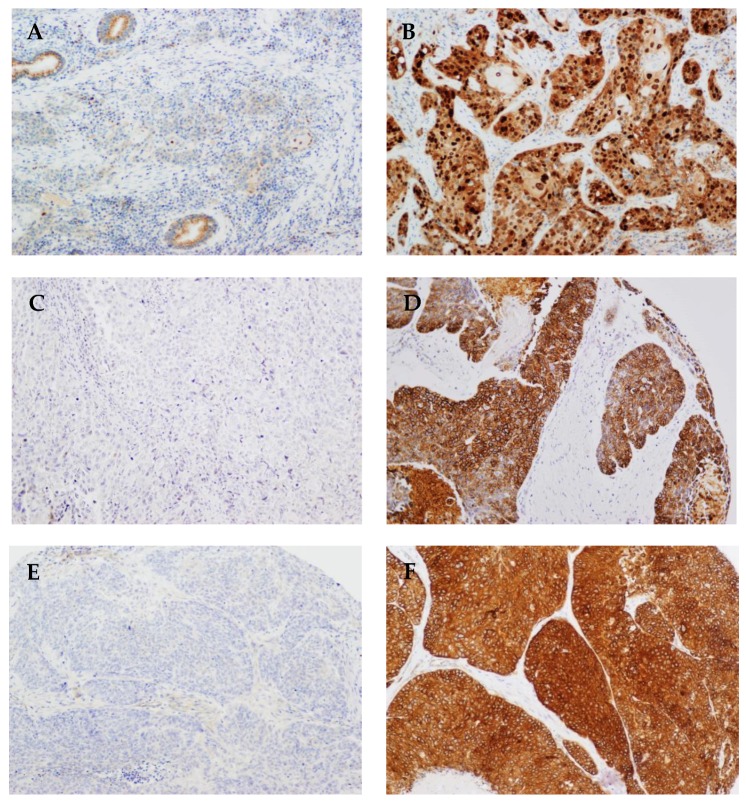
Representative examples of negative and strong positive staining for CCND1 (**A**,**B**), ANO1 (**C**,**D**), and CTTN (**E**,**F**). Original magnification ×20.

**Figure 3 jcm-07-00501-f003:**
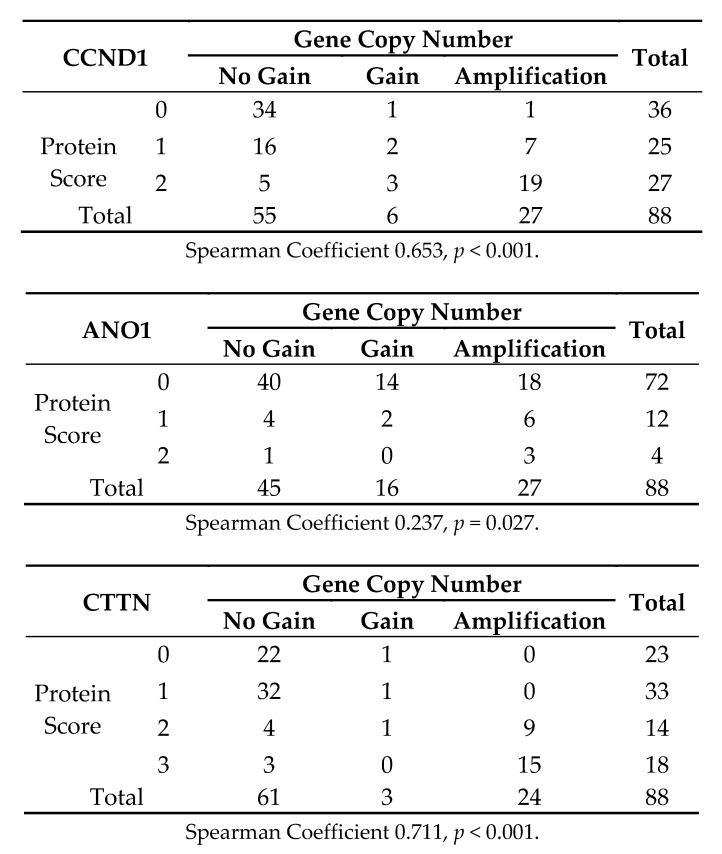
Crosstab to evaluate the correlations between gene gains and amplification and protein staining scores for CCND1, ANO1, and CTTN.

**Figure 4 jcm-07-00501-f004:**
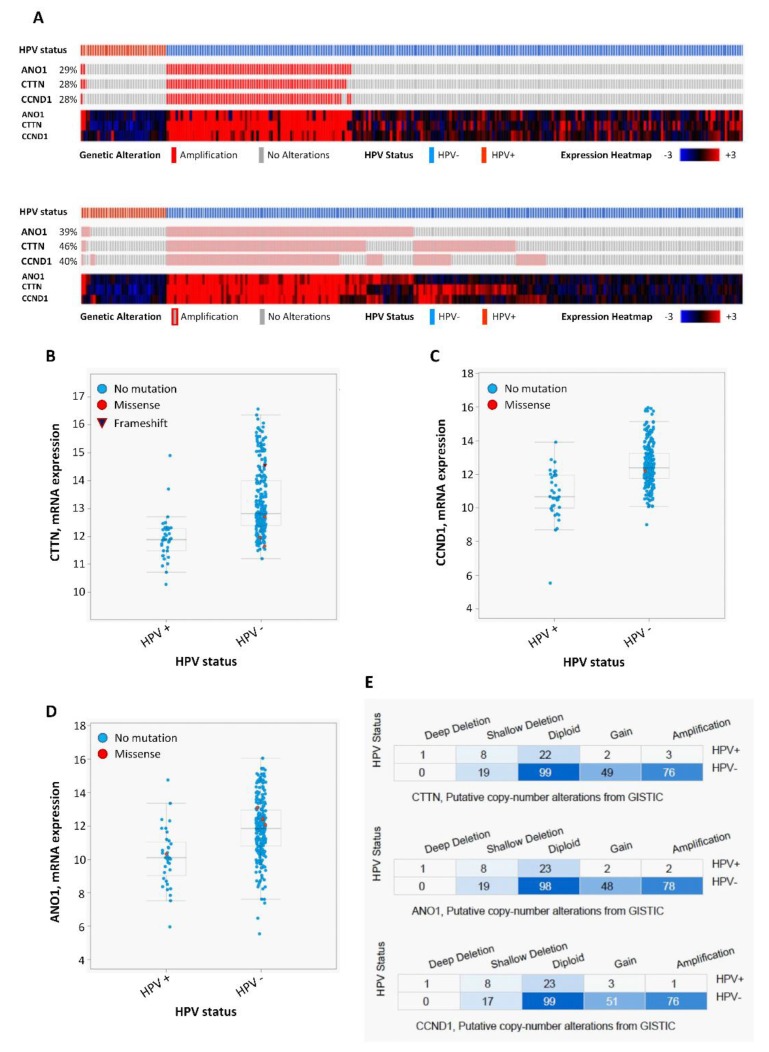
Analysis of mRNA expression and copy number alterations of *CTTN*, *CCND1* and *ANO1* in relation to HPV status related to the available HNSCC TCGA data (*n* = 279) obtained from cBioPortal. (**A**) Schematic representation and heat map showing the percentage of cases with *CTTN, CCND1*, and *ANO1* gene amplification and mRNA expression in relation to the HPV status. (**B**) CTTN, (**C**) CCND1, and (**D**) ANO1 mRNA expression distributed according to the HPV status. mRNA expression (RNA seq V2 RSEM) values were Log2 transformed (*y*-axis). Whiskers plot (min. to max.) with median values; *p* < 0.001, two-tailed Student *t*-test. (**E**) Copy number alterations of *CTTN*, *CCND1* and *ANO1* according to the HPV status using the GISTIC method.

**Figure 5 jcm-07-00501-f005:**
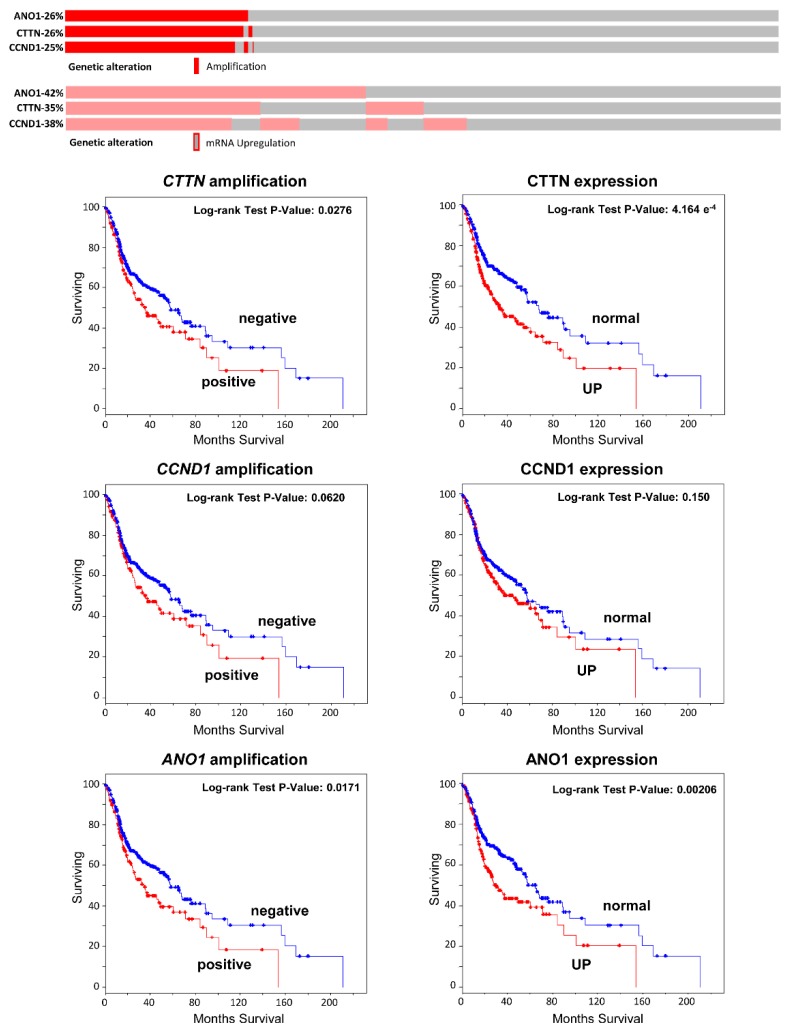
Analysis of *CTTN*, *CCND1* and *ANO1* gene amplification and mRNA expression in the TCGA cohort of 530 HNSCC patients using cBioPortal. Schematic representation showing the percentage of cases with amplification or mRNA upregulation of each gene. Kaplan–Meier survival curves categorized by *CTTN*, *CCND1* and *ANO1* gene amplification dichotomized as positive *versus* negative; CTTN, CCND1 and ANO1 mRNA expression (RNA seq V2 RSEM, *z*-score threshold ±2) dichotomized as normal versus upregulation (UP); *p* values estimated using the Log-rank test.

**Table 1 jcm-07-00501-t001:** Analysis of CCND1 (Cyclin D1), ANO1 (Anoctamin-1), and CTTN (Cortactin) protein expression in relation to human papillomavirus (HPV) status and p16 expression in 392 head and neck squamous cell carcinomas (HNSCC) patients.

Molecular Feature	Number	HPV-Positive	*p* ^#^
**CCND1 protein expression** Negative Positive (scores 1-2)	123 267	21 9	(−0.239) <0.001
**ANO1 protein expression** Negative Positive (scores 1-2)	299 78	29 0	(−0.147) <0.001
**CTTN protein expression** Negative Positive (scores 2-3)	199 190	28 2	(−0.239) <0.001
**p16 protein expression** Negative Positive	325 67	0 30	(0.634) <0.001

^#^ Kendall’s tau correlation coefficient with the associated *p* value.

**Table 2 jcm-07-00501-t002:** Analysis of *CCND1, ANO1*, and *CTTN* gene gain and amplification in relation to HPV status in 88 HNSCC patients.

Copy Number Alteration	No.	HPV-Positive	*p* ^#^
***CCND1* gene** Negative Gain (≥2 copies) Amplification (≥4 copies)	55 6 27	24 2 0	(−0.430) <0.001
***ANO1* gene** Negative Gain (≥2 copies) Amplification (≥4 copies)	45 16 27	22 4 0	(−0.472) <0.001
***CTTN* gene** Negative Gain (≥2 copies) Amplification (≥4 copies)	61 3 24	26 0 0	(−0.422) <0.001
Total Cases	88	26	

^#^ Kendall’s tau correlation coefficient with the associated *p* value.

**Table 3 jcm-07-00501-t003:** Analysis of *CCND1*, *ANO1*, and *CTTN* mRNA expression in relation to HPV status in 279 HNSCC TGCA patients.

Molecular Feature	No.	HPV-Positive	*p* ^#^
**CCND1 expression** Normal UP	206 73	35 1	<0.001
**ANO1 expression** Normal UP	207 72	34 2	0.002
**CTTN expression** Normal UP	177 102	34 2	<0.001
Total Cases	279	36	

^#^ Fisher’s exact test.

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
