# Peer review of "Distinctive Expression and Amplification of Genes at 11q13 in Relation to HPV Status with Impact on Survival in Head and Neck Cancer Patients"

_jcm, 2018, doi:10.3390/jcm7120501_

Round 1
Reviewer 1 Report
In this manuscript, the authors analyze 11q13 genomic amplification/gain and ANO1, CTTN and CCND1 gene and protein expression (TCGA data and IHC, respectively) with respect to the tumor HPV status. In my opinion this paper is not suitable for publication due to the following concerns:
Major points:
- the authors have analyzed protein expression using TMA. However, one major limitation of TMA is tumor heterogeneity. The authors should at least acknowledge this drawback in the discussion. In addition it is not clear from the material and methods section whether stainings were read by at least two independent pathologists. If yes, interobserver variability should be determined.
- in several places in the text, the authors mentioned they studied “the role” of ANO1, CTTN and CCND1. This statement is highly speculative. Given the size of the 11q13 amplicon, it cannot be ruled out that these genes overexpression is only a secondary consequence, and that they are not the true drivers of tumor progression. The authors cannot make that proposition based on the correlations they observe, in the absence of functional data.
- the major concern is that several papers in the literature already provide data about the link between 11q13 gain, ANO1, CCND1, CTTN gene expression, tumor HPV status and patient prognosis. For example, Dixit R et al, Sci Rep 2015 used the TCGA data as well to show that ANO1 gene expression is higher in HPV-negative tumors and has a prognostic impact. So it is not clear what novel ideas/data are brought to the field by this manuscript. In this context, the authors should emphasize more the impact and potential impact of their work.
Author Response
Comments and Suggestions for Authors:
In this manuscript, the authors analyze 11q13 genomic amplification/gain and ANO1, CTTN and CCND1 gene and protein expression (TCGA data and IHC, respectively) with respect to the tumor HPV status. In my opinion this paper is not suitable for publication due to the following concerns:
Major points:
Point 1: the authors have analyzed protein expression using TMA. However, one major limitation of TMA is tumor heterogeneity. The authors should at least acknowledge this drawback in the discussion. In addition it is not clear from the material and methods section whether stainings were read by at least two independent pathologists. If yes, interobserver variability should be determined.
Response 1: We agree. This study limitation is now mentioned in the Discussion (lines 222-227), as recommended. Nevertheless, it is worth to mention that in order to assess the possible influence of tumor heterogeneity, three different representative tumor areas were selected from each tissue block and analyzed in the TMAs. Results showed that these proteins showed homogeneous and highly concordant expression patterns between the three tissue punches from each tumor.
Indeed, immunostainings were scored blinded to clinical data by two independent observers, with a high level of inter-observer concordance (> 95%), as now mentioned in Methods (lines 342-343).
Point 2: in several places in the text, the authors mentioned they studied “the role” of ANO1, CTTN and CCND1. This statement is highly speculative. Given the size of the 11q13 amplicon, it cannot be ruled out that these genes overexpression is only a secondary consequence, and that they are not the true drivers of tumor progression. The authors cannot make that proposition based on the correlations they observe, in the absence of functional data.
Response 2: We agree with the reviewer. Text has been revised and the term “the role of” has been replaced with “the status of”. As mentioned in the Discussion (lines 253-255), several clinical and functional evidences have emerged supporting the involvement of ANO1, CTTN and CCND1 in tumor progression and resistance to radio/chemotherapy (refs 11, 17-21). However, we agree that it cannot be rule out that these genes overexpression may be secondary, and that other genes within the 11q13 amplicon could act as true drivers of HNSCC progression. This argument has now been included in the Discussion (lines 260-262).
Point 3: the major concern is that several papers in the literature already provide data about the link between 11q13 gain, ANO1, CCND1, CTTN gene expression, tumor HPV status and patient prognosis. For example, Dixit R et al, Sci Rep 2015 used the TCGA data as well to show that ANO1 gene expression is higher in HPV-negative tumors and has a prognostic impact. So it is not clear what novel ideas/data are brought to the field by this manuscript. In this context, the authors should emphasize more the impact and potential impact of their work.
Response 3: This manuscript analyzes CTTN, CCND1 and ANO1 protein expression and gene amplification in a large cohort of HPV positive and negative HNSCC patients (n=392). Despite the amplification of chromosomal region 11q13 has been already reported in HNSCC patients, this is the first report showing associations of CTTN and CCND1 with HPV status providing a thorough analysis and consistent data on both protein and mRNA expression as well as gene amplification using two large independent cohorts of >300 HNSCC patients. Furthermore, we also proved that these genes had a major impact on patient prognosis. Even though the recent study by Dixit et al (now included as ref 17 and discussed in lines 245-249) provided the first evidence of a link between ANO1 expression/amplification and HPV status, it is worth noticing that ANO1 protein expression and gene amplification was analyzed in a small series of 64 pharyngeal tumors and IHC evaluation was performed using TMAs. Therefore, our results further and significantly strength and validate these preliminary findings on ANO1 protein expression using a large independent cohort of 392 HNSCC patients, as well as ANO1 mRNA levels in the TCGA cohort of 530 HNSCC patients. Altogether, the herein presented findings represent advancement through the comprehension of molecular mechanisms behind the different clinical outcome between HPV+ and HPV- HNSCC patients, as stated by Reviewer #3.
Reviewer 2 Report
General Comments
General:
In this paper, the specific role of CTTN, CCND1 and ANO1 genes mapping at 11q13 amplicon in relation to human papillomavirus (HPV) status in 392 head and neck squamous cell carcinomas (HNSCC) patients was investigated. In this work, the authors studied in two large independent cohorts of HNSCC patients. They have found that HPV- related and unrelated tumors are two different entities; hence, they consider that the management of HNSCC patients should move to a more personalized approach. The topic of the manuscript is interested and the reviewer only has some suggestions as follows:
Specific comments
1. In results, the reviewer suggested that the results could be arranged by using hierarchical clustering, and the specific role of CTTN, CCND1 and ANO1 in HNSCC might be understand clearly.
2. In methods, please provide the more detail procedure for tissue microarray construction and DNA extraction.
3. Please provide the results from tissue microarray, provide the schematic drawing of experimental setup, and how to compare the tissue microarray and PCR results.
Author Response
Comments and Suggestions for Authors:
General Comments
In this paper, the specific role of CTTN, CCND1 and ANO1 genes mapping at 11q13 amplicon in relation to human papillomavirus (HPV) status in 392 head and neck squamous cell carcinomas (HNSCC) patients was investigated. In this work, the authors studied in two large independent cohorts of HNSCC patients. They have found that HPV- related and unrelated tumors are two different entities; hence, they consider that the management of HNSCC patients should move to a more personalized approach. The topic of the manuscript is interested and the reviewer only has some suggestions as follows:
Response: We thank the reviewer for the positive comments about our work and findings.
Specific comments
Point 1: In results, the reviewer suggested that the results could be arranged by using hierarchical clustering, and the specific role of CTTN, CCND1 and ANO1 in HNSCC might be understand clearly.
Response 1: Following the reviewer’s suggestion, the results have been rearranged and Figure 4A now includes heat map representations to visualize more clearly data from gene amplification and expression of these three genes in relation to HPV status in the TGCA cohort. Similarly, Tables 1 and 2 also respectively show correlations of CTTN, CCND1 and ANO1 protein expression and gene amplification with the HPV status in our cohort of 392 HNSCC patients.
Point 2: In methods, please provide the more detail procedure for tissue microarray construction and DNA extraction.
Response 2: Further details on the TMA construction and DNA extraction have been included in Methods (lines 284-302), as suggested.
Point 3: Please provide the results from tissue microarray, provide the schematic drawing of experimental setup, and how to compare the tissue microarray and PCR results.
Response 3: A schematic representation of the experimental setup has been included in new Supplementary Figure S1. Please note that previous Supplementary Figures S1 to S3 have been moved to the main text (Figures 1 to 3 in the revised version), according to reviewer 3. Results from the TMAs have been summarized in Table 1 and Figure 3 and representative staining images are now shown in Figures 1 and 2. In addition, raw data of CTTN, CCND1 and ANO1 protein scores in relation to the HPV status for the total cohort of 392 HNSCC patients have now been included in Supplementary Information.
Reviewer 3 Report
This manuscript analyzes CTTN, CCND1 and ANO1 protein expression and gene amplification in a large cohort of HPV positive and negative HNC patients. Despite the amplification of chromosomal region 11q13 has been already reported in HNC patients, this is the first report showing at both protein and mRNA level the alteration of CTTN and CCND1 and, more importantly, their association with the HPV status of HNC patients. Opposite, ANO1 alteration has been recently reported. Therefore, findings presented here represent advancement through the comprehension of molecular mechanisms behind the different clinical outcome between HPV+ and HPV-HNC patients.
The manuscript is well described, methodology and experimental plan are straightforward, and results are clearly reported. My only suggestion is to move Supplementary results in the main text, if there are no space limits.
Author Response
Comments and Suggestions for Authors:
Point 1: This manuscript analyzes CTTN, CCND1 and ANO1 protein expression and gene amplification in a large cohort of HPV positive and negative HNC patients. Despite the amplification of chromosomal region 11q13 has been already reported in HNC patients, this is the first report showing at both protein and mRNA level the alteration of CTTN and CCND1 and, more importantly, their association with the HPV status of HNC patients. Opposite, ANO1 alteration has been recently reported. Therefore, findings presented here represent advancement through the comprehension of molecular mechanisms behind the different clinical outcome between HPV+ and HPV-HNC patients.
Response 1: We thank the reviewer for highlighting the quality and originality of our work. Even though the recent study by Dixit et al (now included as ref 17 and also discussed in lines 245-249) provided the first evidence of a link between ANO1 expression/amplification and HPV status, it is worth noticing that ANO1 protein expression and gene amplification was analyzed in a series of 64 pharyngeal tumors. Therefore, our results further and significantly strength and validate these preliminary findings on ANO1 protein expression using a large independent cohort of 392 HNSCC patients, as well as ANO1 mRNA levels in the TCGA cohort of 530 HNSCC patients.
Point 2: The manuscript is well described, methodology and experimental plan are straightforward, and results are clearly reported. My only suggestion is to move Supplementary results in the main text, if there are no space limits.
Response 2: Supplementary Figures S1 to S3 have been moved to the revised manuscript as Figures 1 to 3, respectively.
Round 2
Reviewer 1 Report
In this revised version of their manuscript the authors have addressed the two first points I had raised.
However, in spite of the fact that the work was carried out on a large cohort and on data from the TCGA, I still feel that this manuscript is confirmatory of previously published data and therefore lacks originality and novelty.
Besides, I strongly disagree with the last comment of the authors : "Altogether, the herein presented findings represent advancement through the comprehension of molecular mechanisms behind the different clinical outcome between HPV+ and HPV- HNSCC patients, as stated by Reviewer #3". The data presented in this manuscript is based on correlations between gene/protein expression and prognosis. Absolutely no conclusion can be drawn about the molecular mechanisms that underlie patients' prognosis.
Author Response
Comments and Suggestions for Authors:
Point 1: In this revised version of their manuscript the authors have addressed the two first points I had raised.
Response 1: We thank the reviewer for the positive comment considering that we adequately responded to the first two points raised.
Point 2: However, in spite of the fact that the work was carried out on a large cohort and on data from the TCGA, I still feel that this manuscript is confirmatory of previously published data and therefore lacks originality and novelty.
Response 2: We truly appreciate the Reviewer’s view of our work but we honestly think that, although a role for ANO1 has already been reported, to our knowledge, this is the first study to assess and demonstrate associations of CTTN and CCND1 with HPV status, thereby performing a thorough analysis and providing consistent data on both protein and mRNA expression as well as gene amplification using two large independent cohorts of >300 HNSCC patients. Furthermore, we also proved that these genes had a major impact on patient prognosis.
Point 3: Besides, I strongly disagree with the last comment of the authors : "Altogether, the herein presented findings represent advancement through the comprehension of molecular mechanisms behind the different clinical outcome between HPV+ and HPV- HNSCC patients, as stated by Reviewer #3". The data presented in this manuscript is based on correlations between gene/protein expression and prognosis. Absolutely no conclusion can be drawn about the molecular mechanisms that underlie patients' prognosis.
Response 3: We apologize for this misunderstanding and having included this sentence in our previous rebuttal letter, as it was not our intention to confront different views from the Reviewers. We agree that no conclusion can strictly be drawn regarding the molecular mechanisms underlying patients’ prognosis. In any case, please note that the conclusions in our study are specifically related to the molecular alterations assessed (i.e. CTTN, CCND1 and ANO1 amplification and overexpression), as well as the associations to HPV status and impact on patient survival (lines 357-361), and no conclusion regarding molecular mechanisms have been included in our manuscript.
Reviewer 2 Report
The authors corrected and added some contents according to reviewer’s suggestions on a point-by-point basis. In view of the scope, novelty, and quality of this work, I would recommend this paper to be published in the Journal of Clinical Medicine.
Author Response
Comments and Suggestions for Authors:
Point 1: The authors corrected and added some contents according to reviewer’s suggestions on a point-by-point basis. In view of the scope, novelty, and quality of this work, I would recommend this paper to be published in the Journal of Clinical Medicine.
Response 1: We are glad to know that all the required corrections have been adequately made and thank the reviewer for recommending our paper to be published in JCM.